# LONG DOCUMENT SUMMARIZATION WITH TOP-DOWN AND BOTTOM-UP INFERENCE

## ABSTRACT

Text summarization aims to condense long documents and retain key information. Critical to the success of a summarization model is the faithful inference of latent representations of words or tokens in the source documents. Most recent models infer the latent representations with a transformer encoder, which is purely bottom-up. Also, self-attention-based inference models face the challenge of quadratic complexity with respect to sequence length. We propose a principled inference framework to improve summarization models on these two aspects. Our framework assumes a hierarchical latent structure of a document where the top-level captures the long range dependency at a coarser time scale and the bottom token level preserves the details. Critically, this hierarchical structure enables token representations to be updated in both a bottom-up and top-down manner. In the bottom-up pass, token representations are inferred with local self-attention to leverage its efficiency. Top-down correction is then applied to allow tokens to capture long-range dependency. We demonstrate the effectiveness of the proposed framework on a diverse set of summarization datasets, including narrative, conversational, scientific documents and news. Our model achieves (1) competitive or better performance on short documents with higher memory and compute efficiency, compared to full attention transformers, and (2) state-of–the-art performance on a wide range of long document summarization benchmarks, compared to recent efficient transformers. We also show that our model can summarize an entire book and achieve competitive performance using $0.27\%$ parameters (464M vs. 175B) and much less training data, compared to a recent GPT-3-based model. These results indicate the general applicability and benefits of the proposed framework.

## 1 INTRODUCTION

Text summarization involves compressing a document and preserving key content and meaning. It can be done in either an extractive or abstractive manner. While an extractive summarization model extracts salient fragments (e.g., words, sentences) from the source document to form a summary, an abstractive summarization system aims to generate a semantically coherent and linguistically fluent summary by conditioning on the document. The abstractive approach aligns better with how a human does summarization and generally performs better than extractive models in recent works (Pilault et al., 2020; Zhang et al., 2020). We thus focuses on abstractive summarization.

The dominant approach for abstractive summarization is to use a Seq2Seq model (Sutskever et al., 2014) with an encoder-decoder architecture instantiated with either RNNs (Hochreiter & Schmidhuber, 1997) or, more recently, transformers (Vaswani et al., 2017). In such a model, an encoder infers the latent representations of observed tokens (words or subwords) in the document, conditioning on which a decoder generates a summary. This paper studies the problem of how to infer good latent representations, which in turn would improve summarization. We propose a framework which (1) assumes a multi-scale latent structure of a document and (2) synergizes bottom-up inference with top-down inference. In a multi-scale structure, high-level variables (like those representing sentences, segments) model the document at a coarser time-scale and abstract away details, and are suitable for capturing long range dependency of the document; in contrast, low-level variables (like those representing tokens) preserves details, and prevent the summary from losing key details. In our framework, the summary is generated by conditioning on token representations (low-level vari-

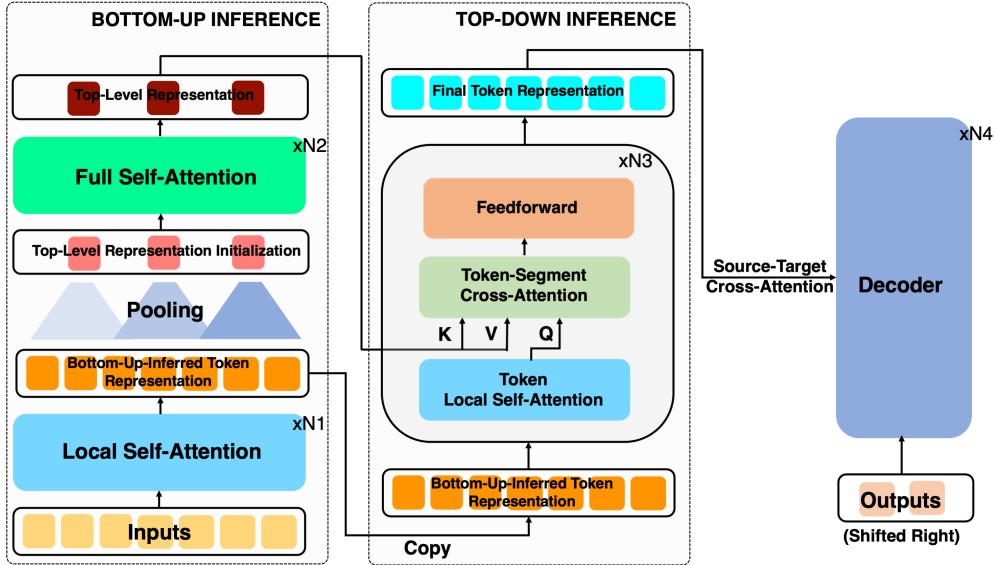

Figure 1: An overview of the top-down transformer. Suppose a document with 7 tokens is the inputs to the model, as shown on the bottom left. The bottom-up inference is achieved with local self-attention ($N_1$ layers) as shown in the left panel. To initialize the top-level representations, we pool bottom-up-inferred token representations with either equal weights or adaptive weights (see Section 2.3 for details). Top-level representations are then updated with full self-attention ($N_2$ layers) to capture global context. They are then used to update bottom-up-inferred token representations, accounting for the top-down update for token representations, as shown in the middle panel. The final token representations are attended by the decoder to generate a summary. Note that inference is used in the sense of statistical inference for latent variables and does not imply no training.

ables), similar to recent abstractive summarization models (Zhang et al., 2020; Zaheer et al., 2020; Beltagy et al., 2020). There is however a critical difference. In our framework, token representations are first bottom-up inferred and then top-down updated with high level representations, hence rendering low-level representations aware of long range information. We hypothesize that the proposed inference approach would improve summarization.

Multi-level models have been widely studied in modeling for images (Sønderby et al., 2016), speech (Mehri et al., 2016), and language (Chung et al., 2016). Prior summarization works (Cheng & Lapata, 2016; Nallapati et al., 2016; Zhang et al., 2019; Xu et al., 2020) have also explored hierarchical models. But they mostly focus on extractive summarization and follow a bottom-up inference approach. They pool information in words or sub-words to form sentence representations, based on which a classification is done to make an extraction decision.

In comparison, our framework combines bottom-up and top-down inference. This draws direct inspiration from a line of work which examines variational inference for hierarchical top-down generative models (Sønderby et al., 2016; Maaløe et al., 2019; Child, 2020). In these models, in the bottom-up path distribution parameters of higher level stochastic variables are computed as a function of lower level stochastic variables, while in the top-down path distribution parameters of lower level variables are corrected for as a function of higher level variables. Although we do not assume stochasticity of the document latent representations, our encoder or inference model follows the same idea to infer token representations.

The proposed framework is agnostic to model architecture. Due to the dominance of transformer models in NLP (Chen et al., 2018; Zhang et al., 2020; Sun et al., 2019; Martin et al., 2020) and to leverage pre-trained language models (Liu et al., 2019; Lewis et al., 2020), we instantiate our framework with a transformer-based model. There is a bottleneck of applying transformers to long documents, because its computational and memory cost has a quadratic dependency on the sequence length. This issue is especially critical for summarization since we are more interested in summarizing long documents since short ones can be quickly read through by humans. To address this issue, a large amount of prior works have been devoted to developing efficient transformers with sub-quadratic complexity. They approach this problem with kernel-based methods (Katharopoulos

et al., 2020; Choromanski et al., 2020), by low-rank approximation to the attention matrix (Wang et al., 2020), by synthesizing the attention weights (Tay et al., 2021), or by designing content-independent (Child et al., 2019; Beltagy et al., 2020; Ainslie et al., 2020; Zaheer et al., 2020) or content-dependent sparse attention mechanisms (Kitaev et al., 2020; Roy et al., 2021; Wang et al., 2021).

Our framework provides a natural way to diminish this quadratic complexity issue. In the bottom-up inference, we use local self-attention where each token only attends tokens within a local fixed-length window, and thus the complexity does not grow as a function of the input sequence length. The top-down correction for the token representations enable them to capture long-range context, reducing the limitation of local attention. Furthermore, in contrast to most prior efficient transformers that are incompatible with pre-training language models, our framework is flexible for leveraging any pre-trained encoder-decoder models such as BART (Lewis et al., 2020), T5 (Raffel et al., 2020).

We call the transformer-based model following the proposed framework as top-down transformer, to emphasize the importance of the top-down inference. We evaluate the top-down transformer on a set of distinct summarization benchmarks. These benchmarks cover documents from a variety of domains, including news articles and scientific, conversational, and narrative documents, and of various lengths ranging from hundreds of words (e.g., a news article), several thousands to over ten thousands of words (e.g., a scientific paper, a book chapter), to even over hundred thousands of words (e.g., an entire book). On short documents, models following our framework achieves on-par or better summarization performance than models with full self-attention, and are more compute- and memory-efficient. Across all long document datasets, our models achieve state-of-the-art performance. In the end, we show that our model is able to summarize a whole book. Compared to a concurrent work (Wu et al., 2021) using GPT-3 and requiring humans to extensively label data, our model achieves competitive performance with 380 times less parameters and a small amount of publicly available data. The diverse and strong empirical results support the effectiveness and wide applicability of the proposed model.

## 2 METHODS

Figure 1 gives a graphical overview of the top-down transformer, instantiating the proposed framework. We introduce its details in this section. Suppose a document has $N$ tokens, $\boldsymbol{t} = \{t_i\}_{i=1}^N$. In our method, token representations are inferred by combining top-down and bottom-up inference. This leads to effective and efficient inference for token representations. They are then attended by a decoder to generate a summary, as in a regular encoder-decoder transformer.

### 2.1 BOTTOM-UP INFERENCE

In the bottom-up inference, contextual embeddings of the tokens, $\{e_i \mid e_i \in \mathbb{R}^d\}_{i=1}^N$, are computed with $N_1$ layers of local self-attention. In particular, each token $t_i$ only attends to nearby tokens within a window of size of $w$. The complexity is hence $O(Nw)$, in contrast to $O(N^2)$ for a full self-attention models. Please see Supplementary B.1 for more details.

### 2.2 TOP-DOWN INFERENCE

The efficiency with local self-attention in the bottom-up inference nevertheless comes with a limitation, that is, each $e_i$ only captures the context within a local window instead of that of the whole document. To mitigate this issue, we propose a top-down inference for token representations.

Consider a two-level multi-scale latent structure for a document. The low level consists of token representations, $\{e_i\}_{i=1}^N$, computed by the bottom-up inference. The top level consists of units at a coarser level. It is affordable to apply full self-attention at the top level due to its coarser granularity, allowing these top-level units to capture global document context. In our work, the self-attention mechanism for the top-level representations is simply the original multi-head self-attention proposed in Vaswani et al. (2017). Readers are referred to Vaswani et al. (2017) for details.

Denote the top level representations after self-attention update as $\{s_j \mid s_j \in \mathbb{R}^d\}_{j=1}^M$ (see Section 2.3 for details on top-level representation initialization methods). We can then update the bottom-up-inferred token representations with the top-level representations. This is achieved with $N_3$ top-down

inference layers, as illustrated by the middle panel in Figure 1. Each layer contains three transformations on $\{e_i\}$: (1) token self-attention, (2) token-segment cross-attention, (3) feed-forward. (1) and (3) are the same as those in the bottom-up inference layers or regular self-attention layer with local attention. (2) implementing the cross-attention between the top and bottom levels is the critical operation. In particular, each $e_i$ is updated with cross-attention,

$$\tilde{e}_i = e_i + \text{LayerNorm}(\sum_{j=1}^{M} \alpha_{ij} f_v(s_j)), \quad \alpha_{ij} = \frac{\exp\left(f_q(e_i)^T f_k(s_j)\right)}{\sqrt{d} \sum_{l=1}^{M} \exp\left(f_q(e_i)^T f_k(s_l)\right)} \quad (1)$$

where $f_q, f_k$, and $f_v$ indicate query, key, and value linear mappings, respectively. For notional clarity, Equation 1 only illustrates the case with a single attention head. In practice, we use multiheads. The cross-attention operation injects global contextual information into bottom-up-inferred token representations, $e_i$, and yields global-context-aware token representations, $\tilde{e}_i$, conditioning on which a summary can be generated by a decoder.

To instantiate the top-down inference, we need to make two choices: (1) the number of top-levels above the token level and (2) the unit representation for each top-level. We choose to use one top level since it is sufficiently coarser to apply full self-attention for a wide range of long document benchmarks we experimented on. A natural choice for top level units is sentence, paragraph, and chapter, depending on the number top level considered. Such a choice however might lead to complicated implementations and non-scalability due to the varying length of these units. We hence choose a simpler approach, where the top level consists of fixed-length segments of the documents. While we use a single top level, multiple top levels can be simply achieved with segments with increasingly coarser granularity.

In the top-down inference, segment-level self-attention has a complexity of $O(M^2)$, and token-segment cross-attention has a complexity of $O(NM)$. Thus, together with bottom-up inference, the complexity is $O(Nw + M^2 + NM)$. In practice, we use relatively small $w$ (window size) and $M$ (number of segments).

## 2.3 POOLING METHODS

As aforementioned, we use a single top level, consisting of fixed-length segments, in the current work. The segment representations are initialized by pooling token representations. Following the notation above, suppose a document is divided into $M$ segments, and the embedding of the $j$th segment is initialized as,

$$s_j^{(0)} = \sum_{n=1}^{k} p_n e_{j \times d + n} \quad (2)$$

where $k$ is the kernel size and $d$ is the stride. $p_n$ is the weight for the $n$th token. We introduce two approaches to compute the weights. The first method is average pooling (AvgPool) and hence $p_n = \frac{1}{k}$, which is simple and convenient. In the second approach, we leverage the reference summary to define the importance of each token to assign adaptive weights (AdaPool). Particularly, we learn an importance tagger with labels constructed with the reference summaries, which involves three steps:

1. construct training labels for the importance tagger: (1) word lemmatization for document and reference words; (2) label a document word as important if it appears in the reference word list and is a non-stopword
2. train a top-down transformer encoder with constructed labels as the importance tagger
3. train the summarization model with oracle weights (i.e., constructed labels from Step 1.) and test it with the adaptive importance weight assigned by the learned tagger

In our experiments, we also used OracleAdaPool where the weights are obtained from Step 1 with the reference summaries. Note that if $\{p_n\}_{n=1}^{k}$ does not form a valid probability distribution, $s_j$ can be computed with a normalized weight distribution within each pooling window as follows,

$$s_j^{(0)} = \frac{\sum_{n=1}^{k} \exp(p_n) e_{j \times d + n}}{\sum_{n=1}^{k} \exp(p_n)}. \quad (3)$$

$\{s_j^{(0)}\}_{j=1}^{M}$ are updated with self-attention, yielding $\{s_j\}_{j=1}^{M}$, which are then used in top-down inference for token representations, as discussed in Section 2.2.

## 3 Experiments

We thoroughly evaluate the proposed framework on distinct summarization datasets. See Table 1 for a summary of datasets used in the current work. Our model is first evaluated on two standard long document summarization benchmarks, PubMed and arXiv (Cohan et al., 2018). It outperforms various efficient transformers and other approaches and achieves state-of-the-art performance. Although we focus on long document summarization, models under our framework is also applicable to shorter documents. We test our model on CNN-Dailymail (See et al., 2017), the most widely used short summarization dataset. Compared to a full self-attention model, our model achieves competitive or better performance but is more memory- and compute-efficient. Recently, a more challenging benchmark, SummScreen (Chen et al., 2021), is proposed, where summarization systems need to summarize TV show scripts. These documents convey plot events often indirectly and implicitly in dialogues, in contrast to news and scientific articles where statements follow a logical order and facts are offered explicitly. Moreover, a typical episode contains multiple subplots that proceed in parallel. Solving this benchmark thus requires a system to draw information from utterances spreading out through the entirety of the input and integrate them to a concise description. Our model outperforms strong baselines on this challenging benchmark by a significant margin. Another challenging dataset, BookSum (Kryściński et al., 2021), is also recently released. It covers books from the literature domain, including stories, plays, and novels. Similar to ScreenSum, it requires integrating plot events from indirectly expressed descriptions. A further challenge is to process long-form texts up to hundreds of pages or over 100,000 words. A model under our framework does well on this challenge, achieving competitive or superior performance compared to a concurrent work (Wu et al., 2021) using GPT-3. While the GPT-3-based model has 175 billion parameters and requires human labelers to extensively write summaries and provide reward information, our model with 464 million parameters is 380 times smaller and merely requires training on relatively minimal data. These results suggest our framework is a generally effectively for documents of various lengths, domains.

| Dataset | # Docs. | # Input Words | # Summary Words | Domain |
|---|---|---|---|---|
| PubMed | 133K | 3,224 | 214 | Scientific |
| arXiv | 215K | 6,913 | 292 | Scientific |
| TVMegaSite | 22.5K | 6,420 | 380 | Conversational |
| ForeverDreaming | 4.3K | 7,605 | 113 | Conversational |
| BookSum-Chapter-Level | 12K | 5,102 | 505 | Narrative |
| BookSum-Book-Level | 436 | 112,885 | 1,167 | Narrative |
| CNN-DM | 311K | 906 | 63 | News |

Table 1: Summarization Datasets. It shows the total number of documents, the average number of input words, the average number of summary words, and the domain for each dataset.

We use the same encoder-decoder architecture for all datasets. The encoder has 8 bottom-up inference layers and 4 top-down inference layers for tokens, and 2 self-attention layers for segments. The decoder has 12 layers. The encoder layers for tokens (12 layers) and the decoder layers are all initialized from BART (Lewis et al., 2020) except the parameters for token-segment cross-attention in the top-down inference layers, which are randomly initialized. The self-attention parameters for segments are also randomly initialized. The window size is 1024 unless otherwise specified. Our settings closely follow Longformer (Beltagy et al., 2020) which has 12 layers for the encoder and decoder, is initialized from BART, and uses a local window size of 1024. Thus, comparison with Longformer is a test of the effect of top-down correction for token representations. PubMed, arXiv, and CNN-DailyMail are obtained from Huggingface Datasets [1]. SummScreen and BookSum are provided by the authors. Standard train/validation/test splits, provided by either Huggingface or the dataset authors, are used for all datasets. Model performance is evaluated with ROUGE scores (Lin, 2004). Reported performance is based on the checkpoint with the best validation R-2 score. Summary samples for each dataset generated by our models are provided in the appendix.

### 3.1 Scientific Documents

We first test the effectiveness of our framework on two widely used datasets based on scientific documents, PubMed and arXiv. They consists of long documents of length ranging from several

---

[1]https://huggingface.co/datasets

thousands of words to over ten thousands words. Each document in PubMed is a scientific article, collected from PubMed.com, and the reference summary is the associated abstract. Documents in arXiv are collected from arxiv.org. Three variants of our model with various pooling weights are presented. AvgPool, AdaPool, and OracleAdaPool in Table 2 indicate average pooling, pooling with adaptive weights, pooling with adaptive weights determined by references, respectively.

| | | PubMed | | | arXiv | | |
|---|---|---|---|---|---|---|---|
| | | R-1 | R-2 | R-L | R-1 | R-2 | R-L |
| | Pegasus (568M) | 44.21 | 16.95 | 38.83 | 44.21 | 16.95 | 38.83 |
| | Dancer | 46.34 | 19.97 | 42.42 | 45.01 | 17.60 | 40.56 |
| | TLM-I+E | 42.13 | 16.27 | 39.21 | 41.62 | 14.69 | 38.03 |
| | SSN-DM | 46.73 | 21.00 | 34.10 | 44.90 | 19.06 | 32.77 |
| | BigBird (577M) | 46.32 | 20.65 | 42.33 | 46.63 | 19.02 | 41.77 |
| | Longformer (460M) | 46.97 | 20.23 | 42.88 | 46.63 | 19.62 | 41.83 |
| | LSH | 48.12 | 21.06 | 42.72 | - | - | - |
| **Ours** | Top Down Transformer (AvgPool) (464M) | 48.34 | 21.40 | 44.22 | 48.67 | 20.70 | 43.91 |
| | Top Down Transformer (AdaPool) (464M) | **51.05** | **23.26** | **46.47** | **50.95** | **21.93** | **45.61** |
| | Top Down Transformer (OracleAdaPool) | 55.15 | 26.55 | 50.25 | 64.16 | 33.39 | 56.88 |

Table 2: Results on Scientific Articles. Best performance (not relying on oracle) is in bold, and the second best is underlined.

The experiment results are displayed in Table 2. Pegasus (Zhang et al., 2020) is pretrained on a large-scale of dataset with a pretraining objective specifically designed for summarization. It uses a full self-attention encoder and thus has to truncate the source document due to the quadratic memory complexity. The summarization-oriented large-scale pre-training makes it a strong baseline. Dancer (Gidiotis & Tsoumakas, 2020) takes a divide-and-conquer approach in which the summary is divided into sections and each section is paired to the appropriate section of the document and the model is trained on short sequences and has a low memory requirement. This is a straightforward approach achieving strong performance.

TLM-I+E (Pilault et al., 2020) first extracts salient sentences and then uses a GPT-style model to generate a summary by conditioning on the introduction section and extracted sentences (instead of the whole document), thus reducing memory requirement. SSN-DM (Cui & Hu, 2021) is an extractive model and uses a sliding encoder to process segments of a document and a memory module to capture autoregressive dependency between segments. These two models bear similarities to our model in that they use a multi-scale structure. The extracted only salient sentences in TLM-I+E can be considered a representation of the document at a coarser granularity since salient information is retained. Instead of keeping the coarser representations in the latent space, TLM-I+E reads out them to the observed word space. In SSN-DM, the fixed-size memory module pooling information from each segments can also be considered a high level representation of the document. Despite these similarities, our model, following a principled framework to synergize bottom-up and top-down inference, clearly outperforms these prior models.

BigBird (Zaheer et al., 2020), Longformer (Beltagy et al., 2020), and LSH (Kitaev et al., 2020; Huang et al., 2021) are efficient transformers. BigBird based on Pegasus pre-training combines local attention, random attention tokens, and global attention tokens. LSH uses content-dependent sparse attention based on local sensitivity hashing. Longformer is closely related to our models. It uses the same local attention as in our bottom-up inference except it has an extra [CLS] token which is a global attention token. Longformer is also initialized from BART, same as ours. The only difference is that our models infer token representations with both top-down and bottom-up inference, in contrary to pure bottom-up inference in Longformer. The clear performance improvement over Longformer and other efficient transformers indicates the effectiveness of the synergy of bottom-up and top-down inference.

## 3.2 SHORT DOCUMENTS

To demonstrate the general applicability of the proposed framework, we show its efficiency and effectiveness on short document summarization and compare it to full self-attention inference model. We hypothesize that although the bottom-up inference uses local self-attention (for efficiency), the top-down correction would enable the effectiveness of our inference and hence lead to competitive or better summarization performance.

| | CNN-DailyMail | | |
|---|---|---|---|
| | R-1 | R-2 | R-L |
| BART (Reported) | 44.15 | 21.28 | 40.90 |
| BART (Re-eval) | 43.93 | 20.81 | 40.79 |
| **Ours** Top Down Transformer (AvgPool) | 44.32 | 21.03 | **41.40** |
| Top Down Transformer (AdaPool) | **44.85** | **21.31** | 41.15 |
| Top Down Transformer (OracleAdaPool) | 63.87 | 38.42 | 59.10 |

Table 3: Results on CNN-DailyMail. Best performance (not relying on oracle) is in bold, and the second best is underlined.

Our model parameters are initialized from BART. Hence, BART with full self-attention forms a natural baseline, allowing for direct comparison. In the bottom-up inference, the local attention window size is 256. As shown in Table 3, models under our framework achieve slightly better performance, especially in terms of R-1 and R-L, than BART. It confirms our hypothesis that a synergy of bottom-up inference with local attention and top-down inference with global attention is effective and achieves on-par or better performance as full self-attention.

## 3.3 SUMMSCREEN

| | | TVMegaSite | | | ForeverDreaming | | |
|---|---|---|---|---|---|---|---|
| | | R-1 | R-2 | R-L | R-1 | R-2 | R-L |
| | Extractive Oracle | 49.0 | 11.6 | 46.9 | 38.8 | 11.5 | 33.9 |
| | Longformer | 42.9 | 11.9 | 41.6 | 25.9 | 4.2 | 23.8 |
| | Hybrid (BART + Content Selection) | 38.8 | 10.2 | 36.9 | 25.3 | 3.9 | 23.1 |
| | Hybrid (BART + Oracle Content Selection) | 42.1 | 11.9 | 40.9 | 26.4 | 5.0 | 23.3 |
| **Ours** | Top Down Transformer (AvgPool) | 49.30 | 14.35 | 47.45 | 35.84 | 8.86 | 30.62 |
| | Top Down Transformer (AdaPool) | **51.02** | **14.66** | **49.01** | **36.84** | **9.19** | **31.12** |
| | Top Down Transformer (OracleAdaPool) | 53.55 | 15.63 | 51.29 | 39.54 | 10.08 | 33.59 |

Table 4: Results on SummScreen. Best performance (not relying on oracle) is in bold, and the second best is underlined.

Scientific and news articles often require that facts are offered explicitly and statements follow a logical order, which might allow summarization models to exploit layout and stylistic biases. We next test the proposed framework on a more challenging dataset, SummScreen, which requires a model to draw and integrate information from indirect expressions across a wide range of the document. SummScreen (Chen et al., 2021) provides two datasets, TVMegaSite and ForeverDreaming, collecting from two different TV show transcript websites. Each document is the transcript of a TV show episode and the summary is an associated recap.

Table 4 summarizes the results. Extractive oracle is an extractive method by extracting nearest neighbors based on Rouge scores. Longformer is an abstractive method and takes the whole document as input. Hybrid models first select salient sentences and then input them to BART. Our models outperform these strong baselines and even achieves comparable or superior performance than those having access to oracle information.

## 3.4 BOOKSUM

BookSum (Kryściński et al., 2021) is another challenging dataset, consisting of books from the literature domain including stories, plays and novels. It includes examples on three levels of granularity with increasing difficulty: (1) paragraph-level with inputs with hundreds of words, (2) chapter-level, with inputs with several thousands or over ten thousands of words, (3) book-level, with inputs spanning up to hundreds of pages and over hundred thousands of words. The chapter-level examples have comparable lengths to other popular long-form summarization datasets such as PubMed, arXiv. We first test our models on the chapter level. The book-level summarization is extremely challenging. First, the number of examples (313 books) is limited. Second, a book is too long to fit in current models. We train our model in a curriculum and recursive way to address the two issues.

| BookSum Chapter Level | | | |
|---|---|---|---|
| | R-1 | R-2 | R-L |
| Extractive Oracle | 42.68 | 9.66 | 21.33 |
| BART (406M) | 37.09 | 8.23 | 15.37 |
| T5 (738M) | 37.38 | 8.42 | 16.77 |
| Pegasus (568M) | 36.17 | 7.79 | 16.09 |
| Longformer (460M) | 32.84 | 7.45 | 14.59 |
| BigBird (577M) | 31.78 | 6.50 | 14.17 |
| **Ours** Top Down Transformer (AvgPool) (464M) | 37.99 | 9.10 | 18.02 |
| Top Down Transformer (AdaPool) (464M) | **38.34** | **9.19** | **18.08** |
| Top Down Transformer (OracleAdaPool) | 41.10 | 9.49 | 19.19 |

Table 5: Results on BookSum Chapter Level. Best performance (not relying on oracle) is in bold, and the second best is underlined.

### 3.4.1 CHAPTER LEVEL

Table 5 displays the results. Kryściński et al. (2021) takes a divide-and-conquer approach to summarize chapters. They finetune BART, T5, and Pegasus on the paragraph level data and the chapter summary is obtained by concatenating the paragraph summary. This might miss the intra-paragraph context. Our models directly summarize the whole chapters and outperform these divide-and-conquer models. Efficient transformers, Longformer and BigBird, are also able to take in the whole chapters as inputs. But these bottom-up approaches clearly underperform our models.

### 3.4.2 BOOK LEVEL

We first train a top-down transformer on the chapter-level data and then fine-tune it on the book-level data. The inputs to the book-level model are (1) the concatenated chapter reference summaries in training or (2) the concatenated chapter summaries generated by the chapter-level model in testing. The chapter-to-book curriculum training is to mitigate the scarcity of book-level data. The recursive summarization of chapters and then books can be considered abstractive content selection applied to book data, and is used to address the extremely long length of books.

| BookSum Book Level | | | |
|---|---|---|---|
| | R-1 | R-2 | R-L |
| Extractive Oracle | 46.62 | 9.17 | 18.31 |
| BART | 29.97 | 6.02 | 10.97 |
| T5 | 39.46 | 7.69 | 13.77 |
| Pegasus | 35.29 | 6.79 | 12.71 |
| 175B full tree RL | 41.51 | 10.46 | 16.88 |
| 175B first subtree RL | 43.19 | 10.63 | **17.10** |
| 6B full tree RL | 36.79 | 7.22 | 14.84 |
| Top Down Transformer (464M) | **44.19** | **10.89** | 16.13 |

Table 6: Results on BookSum Book Level. Best performance (not relying on oracle) is in bold, and the second best is underlined.

Table 6 summarizes the book-level results. The middle section shows the performance for the models with the divide-and-conquer approach (Kryściński et al., 2021), same as those for the chapter-level data. A concurrent work (Wu et al., 2021) based on GPT-3 with reinforcement learning (RL) also attempts to summarize books. Their method shares similarity with ours in that they decompose books into shorter sequences and train the model and summarize the text segments recursively. There are four major differences between our approach and theirs. First, our model has only 464 million parameters and is 380 times smaller than GPT-3 with 175 billion parameters. Second, we train our model with the limited and publicly available data from BookSum, while Wu et al. (2021) requires human labelers to write summaries and give preference, which is highly costly. Third, our model has lower complexity, allowing it to takes in longer input. Thus, we only need to decompose the book one time (into chapters), in contrast to multiple recursive decomposition steps. Multiple recursive summarization steps is prone to accumulating errors. Forth, GPT-3 uses bottom-up inference to infer token representations, in contrast to the synergy of bottom-up and top-down inference in our approach, which we believe leads to better representation inference. The last two differences might account for our competitive performance using a much smaller model and less data.

## 4 RELATED WORK

**Summarization Models**  Prior works have proposed extractive models (Nallapati et al., 2017; Cui & Hu, 2021), abstractive models (Nallapati et al., 2016; Zhang et al., 2020), and hybrid models combining extractive and abstractive methods (Gehrmann et al., 2018; Pilault et al., 2020), for text summarization. Although our model mostly follows the abstractive approach, it also has connections to the hybrid models. These models usually first extract salient sentences from the source document and then summarize the extracted sentences with an abstractive model. Extracted sentences can be viewed a high level representation of the document, although it is the observed space but not in the latent space as in our framework. A continuous representations in the latent space facilities end-to-end learning. Moreover, assigning importance weight with the importance tagger in our method resembles an extractive step in a hybrid model, and thus top down transformer with learned importance tagger can be considered a hybrid model.

**Inference for Hierarchical Latent Variable Model**  Our work draws inspiration from the latent variable inference for hierarchical top-down generative models. To faithfully infer multi-layer latent variables needs to account for the dependency between them. MCMC approaches Nijkamp et al. (2020) naturally accounts for such dependency. Amortized inference Sønderby et al. (2016); Maaløe et al. (2019); Child (2020) makes a special design to capture the multi-layer dependency. In particular, in a bottom-up path, the parameters of the distribution of higher level variables are computed as a function of the lower level variables; in a top-down path, the parameters of the distribution of lower level variables are corrected as a function of the higher level variables.

**Efficient Transformers**  Despite the effectiveness of transformers on a variety of tasks, its quadratic complexity with respect to the sequence length has limited its application to problems with long sequences. A large amount of works have attempted to address this limitation. A major line of work focuses on designing various sparse attention mechanisms. These works can be roughly categorized into two groups, depending on whether the sparsity pattern is content-dependent (Kitaev et al., 2020; Roy et al., 2021; Wang et al., 2021) or content-independent (Child et al., 2019; Beltagy et al., 2020; Ainslie et al., 2020; Zaheer et al., 2020). Our work is mostly related to content-independent sparse attetntion. A main assumption of content-independent sparse attention is that the context temporally and/or spatially proximate to the query token is more important, which is intuitively sensible and supported by empirical attention analysis (Child et al., 2019). Thus, a common and basic sparse attention pattern is local attention, where each query token only attends to a neighborhood within a fixed temporal and/or spatial window. While this reduces the complexity to be linear, a model with only local attention cannot model long-range dependency. Prior works combine local attention with other attention patterns with wider or global receptive field such as dilated attention, random attention tokens, and global attention tokens (Beltagy et al., 2020; Zaheer et al., 2020). Our models also use local attention for its efficiency and leverage top-down inference to enable global-context awareness.

## 5 CONCLUSION

In this work, we propose a principled inference framework to improve latent representation inference for summarization models. It assumes a hierarchical latent structure of a document where the top-level captures the long range dependency at a coarser granularity and the bottom token level preserves the details. We leverage this hierarchical structure and synergize bottom-up inference with top-down inference to improve token representation inference. In the bottom-up pass, token representations are inferred with local self-attention to exploit its efficiency. Top-down correction is then applied to allow tokens to capture long-range dependency. We demonstrate the effectiveness of the proposed framework on a wide range of summarization datasets, including narrative, conversational, scientific documents and news. Our model achieves (1) comparable or superior performance on short documents with higher memory and compute efficiency, compared to full attention transformers, (2) state-of–the-art performance on a wide range of long document summarization benchmarks, compared to recent efficient transformers, and (3) competitive performance on summarizing whole books using $0.27\%$ parameters and much less training data, compared to a recent GPT-3-based model. These results indicate the general applicability and benefits of the proposed framework.

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

# A  QUALITATIVE EXAMPLES

---

**PubMed Example #1: Reference**

---

a new class of water - soluble c60 transfecting agents has been prepared using hirschbingel chemistry and assessed for their ability to act as gene - delivery vectors in vitro. in an effort to elucidate the relationship between the hydrophobicity of the fullerene core, the hydrophilicity of the water - solubilizing groups, and the overall charge state of the c60 vectors in gene delivery and expression, several different c60 derivatives were synthesized to yield either positively charged, negatively charged, or neutral chemical functionalities under physiological conditions. these fullerene derivatives were then tested for their ability to transfect cells grown in culture with dna carrying the green fluorescent protein ( gfp ) reporter gene. statistically significant expression of gfp was observed for all forms of the c60 derivatives when used as dna vectors and compared to the ability of naked dna alone to transfect cells. however, efficient in vitro transfection was only achieved with the two positively charged c60 derivatives, namely, an octa - amino derivatized c60 and a dodeca - amino derivatized c60 vector. all c60 vectors showed an increase in toxicity in a dose - dependent manner. increased levels of cellular toxicity were observed for positively charged c60 vectors relative to the negatively charged and neutral vectors. structural analyses using dynamic light scattering and optical microscopy offered further insights into possible correlations between the various derivatized c60 compounds, the c60 vector / dna complexes, their physical attributes ( aggregation, charge ) and their transfection efficiencies. recently, similar gd@c60 - based compounds have demonstrated potential as advanced contrast agents for magnetic resonance imaging ( mri ). thus, the successful demonstration of intracellular dna uptake, intracellular transport, and gene expression from dna using c60 vectors suggests the possibility of developing analogous gd@c60 - based vectors to serve simultaneously as both therapeutic and diagnostic agents.

---

**PubMed Example #1: Model Output**

---

a new class of water - soluble c60 transfecting agents with positively charged, negatively charged, or neutral chemical functionalities under physiological conditions was prepared using hirschbingel chemistry. transfection, cytotoxicity, and biophysical structure / activity studies were performed in an effort to elucidate the relationship between the hydrophobicity of the fullerene core, hydrophilicity, and the overall charge state of the c60 vectors in gene delivery / expression. only two positively charged c60 derivatives, namely, an octa - amino derivatized c60 and a dodeca - amino derived c60 vector, showed efficient in vitro dna delivery into cells. increased levels of cellular toxicity were observed for positively charged compounds relative to the negatively charged and neutral vectors, and aggregation was the major factor that negatively affected the cytotoxic profiles of these c60 - vector / dna complexes. the successful demonstration of intracellular dna uptake and gene expression from dna dna conjugates formed from c60 compounds suggests the possibility of developing analogous gd@c60 vectors to serve as both therapeutic and diagnostic agents.

---

**PubMed Example #2: Reference**

---

context : isolated involvement of the appendix in crohn's disease is reported to be 0.2 % to 1.8 %, and is usually associated with ileocaecal crohn's disease in 25 % of ileal and 50 % of caecal disease. while appendicitis in a patient who was previously diagnosed to have ileocaecal crohn's may be managed with appendicectomy and ileocaecal resection, appendicectomy alone when performed for appendicitis in a patient with unsuspected ileocaecal crohn's disease could lead to postoperative complications including enterocutaneous fistula.case report : a young female patient who underwent appendicectomy elsewhere for acute appendicitis presented to us with a persistent enterocutaneous fistula of 6 weeks duration. she had complained of general ill health and occasional altered bowel habits for 6 months prior to the acute appendicitis presentation. our investigations, including a ct scan, suggested the possibility of ileocaecal crohn's disease. she underwent excision of the enterocutaneous fistula and ileocaecal resection, and histopathology of the resected specimen confirmed crohn's disease. in the postoperative period she received mesasalazine. when last seen 2 years later during her regular follow - up, she was found to be in good health.conclusion : the possibility of ileocaecal crohn's disease should be considered in patients presenting with unexplained postoperative enterocutaneous fistula following appendicectomy. a high index of clinical suspicion is required to make a prompt diagnosis and institute appropriate further treatment in form of ileocaecal resection.

---

**PubMed Example #2: Model Output**

---

introduction : isolated involvement of the appendix in crohn's disease is reported to be 0.2 % to 1.8 %, and is usually associated with ileocaecal crohn's disease in 25 % of ileal and 50 % of caecal disease. while appendicitis in a patient who was previously diagnosed to have ileocecal ileocecal crohn's may be managed with appendicectomies, appendicectomy alone when performed for acute appendiculectomy could lead to postoperative complications including enterocutaneous fistula.case report : a young female patient who underwent appendiclectomy elsewhere presented to us with a persistent enterococcal fistula of 6 weeks duration. on further questioning, she informed us that she had presented to this hospital with a 2 - week history of abdominal pain, confined to the right lower abdomen. she had complained of general ill health and occasional altered bowel habits for 6 months prior to the acute appendectomy presentation. our investigations, including a computerized tomography scan, suggested the possibility of crohns disease. she underwent excision of the enterococcus fistula and resection of the caecum, and histopathology of the resected specimen confirmed the diagnosis. in the postoperative period she received mesasalazine. when last seen 2 years later during her regular follow - up, she was found to be in good health.conclusion : appendicection alone is a safe procedure in patients with suspected crohn ileocoecal involvement. while appendiceal resection will suffice in those who present subacutely and are diagnosed preoperatively by clinical signs, a high index of clinical suspicion is required to make a prompt diagnosis.

---

Table 7: Summary Samples for PubMed

**arXiv Example #1: Reference**

we study experimentally the thermal fluctuations of energy input and dissipation in a harmonic oscillator driven out of equilibrium, and search for fluctuation relations. we study transient evolution from the equilibrium state, together with non equilibrium steady states. fluctuations relations are obtained experimentally for both the work and the heat, for the stationary and transient evolutions. a stationary state fluctuation theorem is verified for the two time prescriptions of the torque. but a transient fluctuation theorem is satisfied for the work given to the system but not for the heat dissipated by the system in the case of linear forcing. experimental observations on the statistical and dynamical properties of the fluctuation of the angle, we derive analytical expressions for the probability density function of the work and the heat. we obtain for the first time an analytic expression of the probability density function of the heat. agreement between experiments and our modeling is excellent.

**arXiv Example #1: Model Output**

we study the fluctuations of energy and dissipation in a thermostated harmonic oscillator driven out of equilibrium by an external deterministic forcing. this oscillator is very well described by a second order langevin equation. using the fluctuation dissipation theorem, we obtain fluctuation relations for the work given to the system by the external torque and for the heat dissipated by the system, by writing the first principle of thermodynamics between the two steady states. from the experimental results, we derive analytical expressions for the probability density functions of the work and the dissipated heat, and for their finite time corrections, both for a forcing linear in time and a forcing sinusoidal in time. from these expressions, we present some analytical derivations of fluctuation theorems based on experimental observations.

**arXiv Example #2: Reference**

the pairing hamiltonian constitutes an important approximation in many- body systems, it is exactly soluble and quantum integrable. on the other hand, the continuum single particle level density ( cspld ) contains information about the continuum energy spectrum. the question whether one can use the hamiltonian with constant pairing strength for correlations in the continuum is still unanswered. in this paper we generalize the richardson exact solution for the pairing hamiltonian including correlations in the continuum. the resonant and non - resonant continuum are included through the cspld. the resonant correlations are made explicit by using the cauchy theorem. low lying states with seniority zero and two are calculated for the even carbon isotopes. we conclude that energy levels can indeed be calculated with constant pairing in the continuum using the cspld. it is found that the nucleus @xmath0c is unbound. the real and complex energy representation of the continuum is developed and their differences are shown. the trajectory of the pair energies in the continuum for the nucleus @xmath1c is shown.

**arXiv Example #2: Model Output**

the exact solution of the richardson pairing hamiltonian is obtained by including the resonant and non resonant continuum through the continuum single particle level density ( cspld ). the gamow states, which appear in the complex energy representation, provide the main contribution from the continuum. the inclusion of the continuum has been used to study the unbound isotope @xmath0c and beyond. it was found that the continuum pairs ( pair energies with positive real components ) converge to the real part of the uncorrelated pair energy and they do not appear in complex conjugate partners. as a result the total energy of the system may be complex. from the exact solutions of the pairing and pairing - like hamiltonians the csmld can be used to investigate many - body correlations.

Table 8: Summary Samples for arXiv

**CNN-DailyMail Example #1: Reference**

Michelle MacLaren is no longer set to direct the first " Wonder Woman " theatrical movie. MacLaren left the project over " creative differences " Movie is currently set for 2017.

**CNN-DailyMail Example #1: Model Output**

CNN confirms that director Michelle MacLaren is leaving the " Wonder Woman " movie. The movie, starring Gal Gadot, is still set for release on June 23, 2017.

**CNN-DailyMail Example #2: Reference**

Andrew Mogni, 20, from Glen Ellyn, Illinois, had only just arrived for a semester program when the incident happened in January. He was flown back to Chicago via air on March 20 but he died on Sunday. Initial police reports indicated the fall was an accident but authorities are investigating the possibility that Mogni was robbed. His cousin claims he was attacked and thrown 40 ft from a bridge.

**CNN-DailyMail Example #2: Model Output**

Andrew Mogni, 20, from Glen Ellyn, Illinois, had only just arrived for a semester program in Italy when the incident happened in January. He was flown back to Chicago via air ambulance on March 20, but he died on Sunday after falling off a 40 ft bridge in Rome in a suspected robbery attack in Rome, police reports indicated the fall was an accident but authorities are investigating the possibility he was robbed.

Table 9: Summary Samples for CNN-DailyMail

**TVMegaSite Example #1: Reference**

Jake meets Tad at ConFusion where Tad is enjoying a salad. Tad doesn't believe David's story as to why he is in Gloucester. Liza joins them and serves Tad with a restraining order to stay away from the bar in Gloucester. Amanda takes Trevor for an exam at the hospital and joins Angie. David also joins them. Erica sits alone in her hotel room when Opal comes to ask if she's seen the documentary on Pine Valley. Ryan stares at a blank television when Emma comes downstairs with Corinna. Emma asks Ryan if he is going to watch the documentary. In Gloucester, Gayle comes in to check on Greenlee. Greenlee tells Gayle that she has to get home. Greenlee clinches her fist as she imagines how it would be if she were home. Erica panics as to what Ryan might have said on the documentary. Erica receives another copy of the documentary that Hayley did of Pine Valley. David asks Angie if read the gift that he gave her. Amanda tells David that she knew that he had lied to her about having patients to see at the hospital and instead had gone to Gloucester. Tad reprimands Jake for wanting to go to Gloucester without telling Amanda. Jake gets up to go back to work and sees Amanda and Trevor. Jake asks her how long she had been standing there. Amanda answers, " Long enough." Madison and Angie discuss Madison's mom. Erica finally gives in and agrees to watch the DVD. Opal is thrilled, but Erica insists that she wants to watch it alone. Ryan visits with DVD in hand and suggests that they watch it together. David meets with Dr. Clayton and tells him about Greenlee. Greenlee dreams of her family and friends back in Pine Valley. Erica and Ryan watch the DVD. Liza lets Madison know that her father had gotten jail time, but would be out of jail within a year. Amanda and Jake discuss David and how Jake doesn't believe that he is really sick. Amanda tells Jake that if her persists in accusing David then she doesn't know how much longer they can go on. Greenlee meets with Dr. Clayton about her surgery. Ryan and Erica kiss. Liza and Tad kiss in his apartment.

**TVMegaSite Example #1: Model Output**

Tad and Jake are at Krystal's. Jake lets Tad know that David is going to Gloucester. Liza walks in and gives Tad a restraining order against him. At the hospital, Angie tells Amanda that she had seen her on Hayley's documentary. David walks up and listens to their conversation. At home, Opal questions Erica if she had watched the documentary on television about Pine Valley. Erica tells Opal that she doesn't want to see the documentary. Ryan gets ready for Emma and Corinna's sleepover. Ryan lets Emma know that he hadn't been able to watch the documentary that Hayley had shown on television. Greenlee dreams that she is back in Pine Valley with her family. Erica awakens from her coma and finds out that Jackson is alive. Jackson tells Greenlee how much he had missed her. Erica lets Opal know that she would like to fall in love again. Opal asks her if she is insecure about Ryan and what is going on with him. Erica gets a visit from a man, who gives her a DVD of the documentary from Hayley. David and Angie argue over the fact that he isn't as sick as he claims to be. Amanda accuses David of lying to her about where he had been the night that she was stabbed. Tad tries to talk Liza out of breaking into David's bar in Gloucester, but she refuses to listen to him. Madison comes into the hospital and tells Angie that she can not go to her father's sentencing. Madison lets Angie know that her father is being sentenced today and she was going to go, but Angie encourages her to go. Jake and Tad try to talk her out of going to the hearing, but Liza insists on going. Jake tries to get Tad to promise that he will not go into Gloucester without Liza's permission. Erica and Opal watch the video of Ryan's confession. Ryan comes to visit Erica and asks her to watch a little TV. David meets with Dr. Clayton about Greenlee's condition. David introduces Greenlee to Clayton. Tad visits Liza at the bar and apologizes to her for putting her in this position. Tad and Liza begin to argue over his interference in her life. David calls Greenlee and tells her that they are going to take her on a tour of the medical facilities in Gloucestershire.

**TVMegaSite Example #2: Reference**

Erica and Ryan are in her office at Fusion kissing when Greenlee walks up to the door and starts to turn the doorknob. David clasps his hand over her mouth to keep her from screaming. At ConFusion, Liza and Tad kiss before they start to dance. Krystal watches then asks Rob to take her back to his place. Jake and Amanda spend a quiet evening at home when there is an incessant knocking on the door. Jake opens the door and Opal lets them know that she read her tea leaves and knows that someone is headed back into their lives. She fears that it is David. Jake and Amanda don't seem to be too concerned by Opal's anxiety attack over her tea leaves. David reminds Greenlee that Dr. Coleman said she could face another surgery. In talking to David, Greenlee realizes that Ryan is with another woman and comes to the conclusion that it is Kendall. At ConFusion, Tad sees David rushing out to his car. Ryan and Erica come clean to the press that they are involved. Liza visits Amanda and tries to soothe her fears that David is back in town. Tad and Jake visit David at Wildwind. Jake promises David that he will be watching him. Ryan and Erica come home to his penthouse and finds things completely out of order. Erica and Ryan make love in front of the fireplace. Greenlee lets herself into Ryan's place and sees him and Erica making love.

**TVMegaSite Example #2: Model Output**

At the hospital, Liza kisses Tad. Krystal walks in and sees them kissing. Liza asks Tad if she can steal him. At Wildwind, Jake and Amanda are in bed with the baby when there is a knock on the door. Jake answers it and it is Opal. Opal tells Jake that she knows that David is coming back to town. Jake assures her that he doesn't know where David is. At Fusion, Greenlee questions David as to what he is doing here. Greenlee demands to see Ryan, but David refuses to let her see Ryan. David tries to get Greenlee to calm down and let him examine her, but Greenlee insists on going up on the roof to talk to Ryan. At Ryan's home, Erica tells Ryan that she is not used to things going so smoothly in their relationship. Jake tells Opal that Wildwind is being sold and a real estate agent is showing it to the public. Jake lets Opal know that he hadn't heard from David in a while. Amanda comes downstairs and tells Jake about Opal's tea leaves giving her a strong feeling that something from their past is coming to town and that David could be there already. Jake asks Amanda if she is all right. Amanda lets Jake know that David had disappeared. At the Confusion bar, Tad tells Liza that he thinks that there is something going on between him and Krystal, but Liza denies it. David examines Greenlee and lets her know that Ryan is with another woman. David offers to take Greenlee back to his place, but she insists on knowing who the woman is before she changes her mind. Erica asks Ryan to take her home for a romantic dinner and a fire in the fireplace. Ryan and Erica arrive home to find a picture of the moon on the table. David tells Greenlee that he is taking her back to Gloucester for another surgery. David lets Greenlee know that Zach and Kendall had left town with Spike and had taken Spike with them. Jake and Tad burst into David's hotel room and find him hanging upside down in the attic. Jake orders David to leave town. David insists that he will leave town, but Jake tells him that they will be watching for him. David leaves the hotel room, but when he opens the door to leave, he finds that Greenlee is gone. Jake calls Amanda and tells her that they have to file a lawsuit against David because he is back in town.

Table 10: Summary Samples for TVMegaSite

**ForeverDreaming Example #1: Reference**

Sydney races to find a cure for Vaughn, but in order to find the antidote, Sydney must make a deal with Sark that could endanger Sloane's life. Meanwhile, Will continues his research for Vaughn and discovers some disturbing inconsistencies involving 20-year - old standardized IQ tests. Sydney finds out that Vaughn has a girlfriend.

**ForeverDreaming Example #1: Model Output**

Sydney must find a way to save Vaughn's life while trying to track down Sark's base of operations in order to obtain the antidote to a deadly virus that Sloane has been infected with. Meanwhile, Vaughn is in critical condition after being exposed to Rambaldi's toxin, and Sydney is forced to make a dangerous deal with Sark in exchange for his aid in getting rid of him. Will meets with a professor to find out if any of the CIA's standardized tests were ever used to recruit American children in the 1980's, and discovers that one in ten thousand first graders could pass the test.

**ForeverDreaming Example #2: Reference**

Half a universe from Earth, the Doctor and Martha answer a distress call from the cargo ship SS Pentallian, which will impact a star in 42 minutes. The ship scooped out part of the star, which is alive, for cheap fuel. The star uses crew members Korwin and Ashton as host bodies, and begins taking out everyone on board. Martha and crew member Riley, while trying to reach the front of the ship, are jettisoned into space in an escape pod by Ashton. The Doctor remagnetises the pod to bring Martha and Riley back. The Doctor begins being taken over by the star, and tells Martha to vent the engines, getting rid of the " sun particles " in the fuel. This causes the engines to start working again, and frees the ship from the star's pull.

**ForeverDreaming Example #2: Model Output**

The Doctor and Martha are trapped on board a space station in the Torajji system, where the crew are trying to prevent the ship from colliding with the sun. The Doctor uses the sonic screwdriver on Martha's mobile phone to activate Universal Roaming Activation, which allows him to travel anywhere in space and time without interference from the ship's control centre. However, the device malfunctions and the ship begins to fall towards the sun, and the Doctor is forced to use the emergency escape pod to escape. The pod, which contains the Doctor, Martha and two other crewmembers, is destroyed by the impact, but the Doctor manages to return to the control centre to try and stop the ship hitting the sun before it does so.

Table 11: Summary Samples for ForeverDreaming

**BookSum Book-Level Example #1: Reference**

At the opening of Act I, it is a cloudy autumn day on a Russian country estate. In the garden, the old nurse Marina stands at the samovar and offers Doctor Astrov something to eat, but he refuses. He complains about the difficulty of his job. Telegin, an impoverished local landowner, sits with them. Voynitsky, known as Vanya, comes out of the house and joins them. He is almost fifty and is weary and irritable. He complains about his brother-in-law, Serebryakov, Serebryakov's young second wife, Helen, and about how their visit has turned the place upside down. Serebryakov, Helen, and Serebryakov's daughter, Sonya, join them for a moment. After they depart, Vanya sighs about Helen's beauty and then complains about how he has toiled his whole life on this estate for the professor and it has come to naught. After Vanya's sister's death, he and Sonya worked here so the professor could continue his studies and his writings, but Vanya has come to see that work as foolish and irrelevant. When Astrov suggests that Vanya is jealous, Vanya laughs that he obviously is, especially as the old, gout-and-rheumatism-ridden man seems to attract beautiful women. Helen ventures outside and tells Astrov his services are not needed for her husband. Mrs. Voynitsky, Vanya's mother and Sonya's grandmother, tells them about a new pamphlet written by a friend in Kharkov. When Vanya sneers that all they do is read pamphlets, she becomes distressed and claims he hates her. Vanya merely says he is old, tired, and frustrated. A laborer arrives and tells Astrov he is wanted at the factory; the doctor bitterly departs, but not before they all discuss how he is very interested in forestry work. Sonya speaks up cheerfully about how Astrov is trying to save the old forest from destruction because forests make people happier. Astrov speaks of how Russians have torn down the forests and destroyed the wildlife: they no longer create, but rather destroy. After Sonya walks Astrov out, Vanya tries to seduce Helen, but she pushes him away. She muses about how Sonya clearly seems to love the doctor but he does not love her back. Helen sighs that she is simply bored and life is too much for her. In Act II, Serebryakov complains to Helen of how he is old and no one respects him. His querulous behavior only annoys Helen, who begs him to stop it. Serebryakov ignores her and bemoans how his life of scholarship seems to be nothing now. Sonya joins them and tells them Serebryakov must see Astrov now; she wants her father to stop behaving like a child. The elderly nurse Marina comforts Serebryakov and leads him out. Helen tells Vanya, who entered the room, that her husband wearies her. Vanya can only lament that everything is over for him and his life was wasted on trivial things. Helen is annoyed and moves to leave, but he bars her way. She accuses him of being drunk, and he admits to it. After Helen sweeps out of the room, Vanya ruminates on what a fool he was not to fall in love with her when she was younger; he once admired the professor, but now he does not. When Astrov returns, he mocks Vanya for having feelings for Helen, but Vanya will not admit it. Astrov leaves to get a drink; Sonya pulls him aside and makes him promise to stop drinking and stop getting her uncle drunk. He agrees. They continue to talk for a moment. He comments that Helen is beautiful but idle and useless. This country life makes people like that, and he despises it; he has been beaten down and sees no light at the end for himself. The peasants are all the same, and educated people are ridiculous. He only likes forests. Sonya compliments him and tries to cheer him up. As he prepares to leave, she asks how he might feel if he were to out that a friend of hers has feelings for him, and he drolly says he cannot love anyone. After she leaves, Sonya feels a surge of happiness though she is not sure why. In Act III, Sonya confesses to Helen that she loves Astrov, and Helen suggests that she say something to see if the doctor loves Sonya too. Sonya gives her permission for Helen to do this. Astrov and Helen meet to ostensibly look at his forestry maps. He discourses volubly on the patterns of deforestation until he sees that Helen is uninterested. Helen insists she is interested but says they should talk about something else. She point-blank asks if he likes Sonya, and he says no. He then moves in to seduce Helen, but she wants none of it. As he tries to kiss her, Vanya enters the room with flowers. Helen is horrified by the situation and begs Vanya to tell her husband that they must leave today. A moment later, Serebryakov and the others enter and Serebryakov announces that he has an idea to sell the estate because he and Helen need to afford a place in the city. This announcement angers Vanya tremendously, and he begins to complain violently about how Serebryakov is a fraud, is uninspired, is thankless, and how he, Vanya, has labored for Serebryakov his whole life and for no reason. He insists this is Sonya's estate. He runs out of the room. Serebryakov is startled by Vanya's outburst. He insists he cannot stay here anymore. Sonya implores him to talk to her uncle and he agrees. He departs, and those in the room hear a gunshot, then another. Helen and Vanya struggle over a revolver as Vanya screeches that he missed the professor. Cursing, he sinks into a chair. In Act IV, Telegin and Marina discuss Serebryakov and Helen's planned departure for that day, then exit the room. Vanya and Astrov come in. Astrov mocks Vanya for his behavior and asks him to return what he stole. Vanya maintains that he is innocent of theft: it is stultifying and useless. When Sonya enters Astrov tells her to tell her uncle to return the bottle of morphia he stole. Sonya turns to her uncle and tearfully asks him for the bottle. He complies. She takes him to make up with her father. Helen enters to tell Astrov goodbye. He tries to seduce her again in a rather lackluster fashion; she kisses him and bids him farewell. Everyone bids goodbye to Serebryakov and Helen, who leave the estate. Sonya and Vanya return to work. Marina suggests Astrov stay for dinner; he refuses and says he must be off. Astrov leaves, and even though Sonya knows he did not love her, she is sad. Vanya, though, claims he is extremely depressed. With both in tears, Sonya comforts her uncle. She tells him that life may be difficult for them now, but the afterlife will be full of peace, love, and rest.

**BookSum Book-Level Example #1: Model Output**

The scene opens in a country house in the Russian countryside. Ivan, a young man, is sitting with his mother, who is knitting him a Christmas stocking. He tells her that he does not want to have any tea, but she insists that he drink it anyway. She asks him how long they have known each other, and he says that it has been ten years since he first met her. He has not had a day of rest since he has known her, and has toiled without repose or a day's freedom ever since. He had a typhoid epidemic in the third week of Lent, and when he returned home, he had to take care of a man who had died of typhoid fever. He then sat down and closed his eyes for a while and wondered if he had killed the man. He wonders if the future generations will forget him, but he is sure that they will not. Ivan tells them that the house is a mess, because the Professor has been up all night writing and doesn't have time to eat or drink. He asks them to wake him up at two in the morning, but when they do not wake him, he thinks that the Professor wants some tea. They all come in from their walk, and as they come in, they talk about how happy they are to be together, and how they are all living in peace and contentment. Ivan is struck by the sight of a beautiful woman, and says that as he looks at her, his heart swells with unbridled happiness. The work of preserving the forest is thrilling, he says, and Dr. Astroff watches over the old woods and sets out new plantations every year. He wants to show them to Helena and Sonia, and Helena asks him if he finds it interesting. Helena is sitting next to him, and tells him to look for Batushka's works in the library the next day. He is worried that he is getting angina from his old age. Helena tries to tell him to make peace with himself, and to stop being so angry with everyone else. But he says he will not listen to her, because he has spent too much time thinking about the past, and now he cannot bear to watch the success of others and to fear death. Helena gets up and walks away from him, sitting down at a distance. She tries to get him to tell her why he doesn't love anyone. He says he is not an idiot, that he can understand her, but that he cannot understand her because she is young and healthy and beautiful and longing for life and he is an old dotard, almost a dead man. She tells him that she can understand him and that he should be able to understand her. Now, she says, she is too old and sick to continue to live in the country, and she has a young daughter and a young wife to care for. She doesn't want to live on the income generated by the estate, so they have to find some way to guarantee that they can live in town on the money generated each year. They decide to sell the woods, because they don't have enough money to do so every year, and they can't afford to buy the woods every year either. They need to find a way to make sure they can afford a certain amount of money to buy a cottage in Finland. They also have to figure out what to do with the rest of the property. The estate is worth ninety-five thousand roubles, and the estate was bought from his uncle, who had a debt of twenty-five. Semyon Lakedemonoff's brother bought the estate from the Lakedemons' uncle. The family ties are what got the estate to be worth so much money. Everyone agrees that the estate is for Sonia's good, and that she should get to keep it. But now that he has gotten old and ill, the time has come for him to dispose of his property in regard to the interests of his VOITSKI, ASTROFF, SEREBRAKOFF, Vanya, and SONIA arrive at the house to say goodbye to Tommo and Marina. They are to move to Kharkov to live with the professor and his wife. They have been frightened by what they have just witnessed, and decide to go to the city to see if they can find a place to live there. They will not be staying in the village any longer. Except for Vanya and Alexander, who stay to say good-bye to his wife and son-in-law. The Professor kisses them all three times, and then goes out to see them off. He gives them one last kiss to each of them before he leaves. They say they will always remember each other with pleasure, that they are interesting and original, and original. They shall rest

Table 12: Summary Samples for BookSum Book-Level

**BookSum Book-Level Example #2: Reference**

In his London studio, artist Basil Hallward puts the finishing touches on his latest portrait, that of a young man. Although Lord Henry, who is visiting with Basil, asks about the young man's identity, Basil declines to answer, noting his preference for secrecy. Basil never intends to exhibit the painting, because if he did, it would bare the deepest feelings in his soul. However, Basil lets slip that the subject of the portrait is Dorian Gray, who shortly thereafter pays the two men a house call. Lord Henry immediately begins to influence Dorian, suggesting that he should treasure and guard his youth and beauty while he has them, because they will soon fade. Terrified of aging, Dorian wishes he could trade his soul to stay as young as he looks in the portrait; a short while later, he again wishes that he could stay young while the image in the painting aged. The portrait thus begins to take on a life-like existence; in fact, Basil's threat to burn the portrait is likened to "murder" and Basil prefers the company of the portrait to the real Dorian. Dorian falls in love with a young actress, Sibyl Vane, a woman he barely knows. She plays a different woman at each night's performance, earning the label of "genius" from Dorian, who is as smitten with her acting more than with her personality. They become engaged, much to the surprise of Lord Henry and Basil. The sweet, wholesome Sibyl discusses her engagement with her family. Because her mother is indebted to the theatre manager, Mr. Isaacs, for fifty pounds, she is against the marriage unless Dorian is wealthy; they do not know that he is. Sibyl's angry brother, James, is leaving for Australia, but he vows to kill Dorian if he wrongs his sister in any way. James also confronts his mother about gossip he has heard – that his mother and deceased father never married, which Mrs. Vane admits is true. Dorian attends a performance of Sibyl's with Lord Henry and Basil, but the performance is terrible. Sibyl tells Dorian she can no longer act, because he has shown her a beautiful reality. Dorian is disgusted by her poor acting, because her performances were what drew him to her; he dismisses her and returns home. To his surprise, the portrait shows marks of cruelty around the mouth, lines that do not show on Dorian's face. He begins to suspect that his wish is coming true, so he vows to be good so that both he and the portrait can remain young. He, therefore, intends to apologize to Sibyl the next day and makes to marry her after all. However, he is too late: Sibyl commits suicide at the theatre that night. Dorian first feels responsibility for her death, but then views it both as wonderful entertainment and a selfish act on her part. Lord Henry tries to keep Dorian's name out of the scandal. Dorian and Lord Henry spend the evening at the opera. The next morning, Basil arrives and expresses concern for Dorian, given the events of the previous day. Dorian, however, is completely unconcerned about Sibyl or her family; he wants to talk only of happy subjects. The next day, he covers his portrait and moves it to the attic, to which Dorian has the only key. He then settles in to read a yellow book sent by Lord Henry; the book becomes Dorian's blueprint for life. Several years pass, and Dorian lives a hedonistic life according to the guidelines established by Lord Henry and the yellow book. While the face in the portrait has turned ugly, Dorian remains young, beautiful, and innocent. People talk about Dorian's "madness of pleasure" and his dreadful influence on the people around him, but that is of no consequence to him. Finally, when he is thirty-eight years old, Dorian shows the portrait to Basil, who begs Dorian to repent of his sin and ask that the wish be revoked. Instead, Dorian kills Basil and hides his body. Blackmailing his old friend Alan Campbell, Dorian is able to dispose of Basil's body. An hour later, Dorian attends a party, but is bored and distracted. He then heads for an opium den and, out on the street, meets Sibyl's younger brother, who has been waiting for an opportunity to harm Dorian for nearly twenty years. Dorian makes a case for mistaken identity when he claims to have the face of a twenty-year-old and cannot be the man James is looking for. A woman in the street reveals that Dorian "sold himself to the devil for a pretty face," so James again pursues Dorian. At his country estate one week later, Dorian entertains guests but believes James in hunting him. Dorian soon learns, however, that a man accidentally killed in a hunting accident is James, and so he feels safe. The novel concludes six months later. Dorian and Lord Henry dine, and talk turns serious – Dorian talks of Basil, and Lord Henry reflects on a sermon he heard the previous Sunday while walking in the park. Lord Henry also inquires about the secret of Dorian's youth, which Dorian dismisses. Dorian then asks Lord Henry never to give the yellow book to anyone else. That evening, while Dorian examines the portrait, he decides to destroy it with the knife used to murder Basil. Soon after, Dorian's servants and a police officer find an old, ugly man lying dead on the ground in front of a portrait of a young and innocent Dorian.

**BookSum Book-Level Example #2: Model Output**

Dorian Gray is sitting in the garden of his uncle's house in London, playing the piano. In the center of the room is a portrait of a young man of extraordinary beauty, and in front of it is Basil Hallward's portrait of Dorian Gray. The portrait belongs to Hallward, and he refuses to show it at the Grosvenor Academy, because it contains too much of himself in it. He tells Dorian's uncle, Lord Henry Wotton, that he is in love with Dorian and wants to marry him. Dorian is sitting at the piano and finishes a painting that he has been working on. He looks at the portrait and realizes that it is of him. He realizes that he will never be able to look at it again, but he cannot stop looking at it because he wants to remember the beauty of his own face. He asks his uncle for some information about Dorian, and his uncle tells him that Dorian has a beautiful mother who was married to a poor man who was killed in a duel. She left him a son, who is very good-looking and who has inherited all of her property. Lord Henry tells him to write to him and ask for some advice, and Dorian agrees. One day, Dorian meets Sibyl Vane, a beautiful young woman who works as a governess for a rich family in the East End of London. She is in the employ of Lord Henry's friend, Mr. Erskine of Treadley, and Lord Henry wants to see her. He also wants to get her out of the hands of the Jew who has her bound to him for three years and eight months. He proposes to her, but she refuses him. She says that she does not think he is good enough for her, and she will never love anyone of his rank. He is disappointed, but does not say anything to his mother about it. The next day, he meets the Duchess of Monmouth, who tells him he should find a wife and marry her. She wants him to have a future and not to spend his money frivolously. He agrees, but when he tells her that he does not love her, she laughs at him and refuses to call him by his new name, Prince Charming. He goes to see the play, and is horrified to see that the face on the canvas is that of the portrait of Romeo and Juliet. He cannot believe that he could have done such a terrible thing to Juliet and that she could still be his wife. He leaves the theater and wanders the streets of London until he finds himself in Covent Garden. He finds some women waiting for him, and one of them laughs when he calls her by his nickname, "Prince Charming." She curses him and runs away. He runs into a dark alley and is suddenly grabbed by a man with a gun pointed at his head. It is James Vane. Vane threatens to kill Dorian if he doesn't make peace with God. He gives Dorian one minute to make his peace before he kills him. When Dorian gets to the street, he finds that the man he was trying to kill is not the same man he thought he was. It turns out that Vane is twenty-eight years younger than Dorian. The woman who took his money tells him not to talk to her again. She runs off, and when Dorian looks back, the woman has disappeared. When he wakes up the next morning, he has not had a nightmare. He writes two letters to his assistant, Alan Campbell, telling him that there is a dead man sitting on a table in his house, and that he must destroy the body so that no one will ever know who he is. He then goes to his bedroom and finds a small box of lacquer, which he takes out and puts inside. He puts the box back, gets into a horse-drawn carriage, and gives the driver an address. The driver takes him to the address, and as he is leaving the house, he sees the dead body of a man on the table. When Campbell returns, he tells Alan not to disturb the body, but to come back at seven o'clock in the evening. When the man arrives, he throws the picture over the table, but Dorian does not believe that it has been disturbed. He returns home and finds that Campbell has brought back the chemicals and the irons, and the other things that he needs to do the job. He opens the cabinet where he had hidden Basil's coat and bag, and finds the green paste. At midnight, he gets a hansom and leaves the house with the instructions to meet him at 7 o' clock the next day. He sits in the back of the carriage as the driver drives him through the streets. He wonders if it is possible to cure the soul by means of the senses and the body by way of the soul. He wakes up in the middle of the night to find that the portrait has not changed.

Table 13: Summary Samples for BookSum Book-Level

## B ADDITIONAL EXPERIMENT DETAILS

Due to the space limit, additional experiment details are reported here. As we discussed in the main text, all of our models are initialized from BART Large with 12 layers (Lewis et al., 2020). In the bottom-up inference module (the left panel in Figure 1), the local self-attention in our models has 8 layers and all parameters are initialized from the first 8 layers of BART (including parameters for layer normalization). We use 4 layers for top-down inference (the middle panel in Figure 1). Each layer consists of (1) token local self-attention, (2) token-segment cross-attention, and (3) feed-forward. (1) and (3) are initialized from the last 4 layers of BART (including parameters for layer normalization). All other parameters are randomly initialized. The segment-pooling has a kernel size of 32 and a stride size of 24. The maximum document lengths for PubMed, arXiv, CNN-DM, TVMegaSite, ForeverDreaming, BookSum are 8192, 16384, 1024, 12288, 12288, 12288, respectively.

### B.1 LOCAL SELF-ATTENTION

The local self-attention used in our work is widely used in prior works on sparse attention (Beltagy et al., 2020; Zaheer et al., 2020). It is illustrated in Figure 2. It shows local self-attention of 9 tokens with window size 4. Each token attends 2 tokens on the left and 2 tokens on the right, as long as there are sufficient right and left neighbors. The attended nearby tokens are in light green. Each token also attends itself, as indicated by dark green. White color in Figure 2 indicates absence of attention.

It is also called sliding window attention (Beltagy et al., 2020). We call it local self-attention to make a direct contrast with the full self-attention used for the segment-level representations. Despite its efficiency, it misses information outside of the local attention window. Thus, it is often used together with other attention mechanisms. In Longformer, sliding window attention is combined with dilated sliding window attention and global token attention. In BigBird, it is combined with random attention and global token attention. In our work, we use segment-level tokens to collect long range information which is then used to enrich token-level representations through token-segment cross attention (see top-down inference in Figure 1).

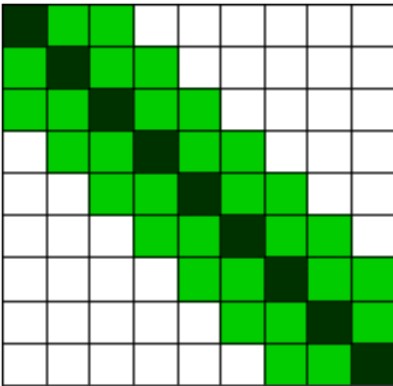

Figure 2: An illustration of local self-attention. It illustrates local self-attention of 9 tokens with window size 4. Each token attends 2 tokens on the left and 2 tokens on the right, as long as there are sufficient right and left neighbors. The attended nearby tokens are in light green. Each token also attends itself, as indicated by dark green. White color indicates absence of attention.

# C  ABLATION STUDIES

We present results for a series of ablation studies in this section. The experiments are performed with PubMed. The results are summarized in Table 14. The first row shows the performance of the top-down transformer with top-down update via cross-attention and window size 1024, which is our final model (Please see Figure 1 for an illustration).

The second row shows the performance for a variant of top-down update. In this variant, to update the bottom-up inferred token representations, we concatenate the token representations with the corresponding top-level segment representations, in contrast to the cross-attention approach used in the final model. We can see a clear performance degradation, indicating the importance of the cross-attention-based top-down update.

The third row displays the results without top-down update, and the decoder attends the bottom-up-inferred token representations to generate summaries. Compared to our final model, the performance is also degraded, suggesting the effectiveness of the top-down update.

The bottom panel of Table 14 presents ablation results on the window size of local self-attention (see Figure 2 for an illustration). These results are also plotted in Figure 3. They show an effect of window size. That is, as the window size increases, the performance on all metrics enhances. The effect is quite large when the window size is increased from 32 to 256. The effect becomes smaller after 256, but the model performance can still benefit from larger window size.

| | | | R-1 | R-2 | R-L |
|---|---|---|---|---|---|
| Top-Down Transformer | with top-down update via cross-attention | window size - 1024 | 48.34 | 21.40 | 44.22 |
| | with top-down update via concat | window size - 1024 | 47.04 | 20.36 | 43.03 |
| | without top-down update | window size - 1024 | 46.97 | 20.23 | 42.88 |
| | with top-down update via cross-attention | window size - 32 | 46.30 | 19.55 | 42.21 |
| | with top-down update via cross-attention | window size - 64 | 47.25 | 20.37 | 43.12 |
| | with top-down update via cross-attention | window size - 128 | 47.44 | 20.56 | 43.35 |
| | with top-down update via cross-attention | window size - 256 | 47.89 | 21.06 | 43.77 |
| | with top-down update via cross-attention | window size - 512 | 48.08 | 21.16 | 44.05 |

Table 14: Ablation studies of Top-Down Transformer with PubMed.

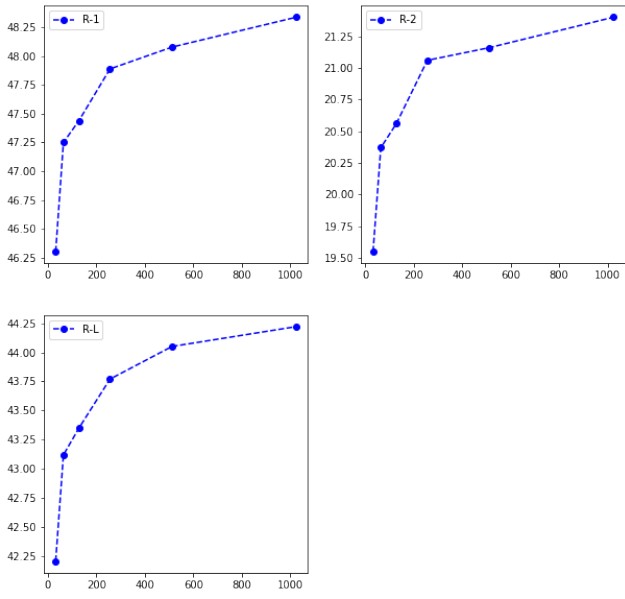

Figure 3: Ablation on attention window size with PubMed. The window sizes tested are 32, 64, 128, 256, 512, 1024.

