# OpenReview forum: "Long Document Summarization with Top-Down and Bottom-Up Representation Inference"
_ICLR.cc/2022/Conference — ICLR 2022 Submitted_

### Official Review · Reviewer_fnCT · 2021-10-31

**Correctness:** 3
**Technical Novelty And Significance:** 2
**Empirical Novelty And Significance:** 3
**Recommendation:** 5
**Confidence:** 3

**Main Review:**

Strengths
1、This paper conducted a variety of experiments on long document data sets of different levels, and proved that the top-down structure can effectively improve the performance of long document summaries.
2、Compared with the model of gpt-3, the model proposed can summarize a complete book, and use 0.27% parameters and less training data to achieve competitive performance.
3、The experimental result has proved that the proposed model can effectively improve the performance of long text summaries, which has been greatly improved comparing with other baseline models.
Weaknesses
1、The method proposed in this paper is not innovative enough. Similar methods have been extensively studied in images and speech, and the model structure does not fully reflect the problem of information redundancy faced by long document summaries.
2、The detailed description of the model in this article is not clear enough, for example, there is no detailed description of the overall operation process of the local attention and the top-down model.
3、In this paper, the experimental verification for calculation efficiency and memory usage of model is not sufficient. It only compares the parameters of the model, and does not conduct quantitative analysis through experiments.

**Summary Of The Paper:**

Aiming at the task of long text summarization, this paper proposes a top-down and bottom-up reasoning method to improve the traditional bottom-up converter encoder structure, which has higher memory and computational efficiency. The model captures remote dependencies on a coarser time scale at the top, and the token level at the bottom retains details. It uses local self-attention to improve computational efficiency. At the same time, it has achieved good results on a large number of long text summary data sets. However, the method proposed in this article has been extensively studied in many fields, and there is no detailed analysis of the problems of long texts, and the theoretical innovation is insufficient.

**Summary Of The Review:**

1、It is recommended to explain the specific calculation process of the model in detail, especially for the top-down inference process;
2、It is recommended to further verify the computational efficiency and memory usage of this method through experiments.

---

> ### Author Response · Authors · 2021-11-20
> **Response to feedback**
>
> Dear Reviewer:
>
> Thank you for your valuable suggestions. Your concerns are addressed in order as follows.
>
> 1. *Regarding relations to prior works:*
>
> We have followed your suggestions and looked for related works. We found two recent papers, MViT [1] and CrossViT [2]. Both papers are built on top of Vision Transformer (ViT) and leverage multi-scale features to improve ViT. Our work shares a core idea with these concurrent works, that is, use of multi-scale features in transformers. This is a rather broad idea and can be instantiated in various distinctive ways to address different problems.
>
> We would like to point out the following distinctions between our work and the two concurrent works.
>
> Data Modality:  [1] and [2] both apply this idea to address classification problems in computer vision, while we are addressing text document summarization, from short documents to regular long documents to extremely long documents (books).
>
> Nature of Problem: Due to the nature of the problems being different, [1] and [2] are an encoder model, while our model is an encoder-decoder model for seq2seq processing.
>
> Information Flow: [1] gradually pools features to reduce the granularity and the classification is done via the coarsest features. [2] uses two parallel branches to process features of different scales, and they share information with each other through two [CLS] tokens (one for each scale), and the classification is done by pooling the representations of the two [CLS] tokens. In our work, since critical details need to be preserved for good summarization, the segment-level representations (at a coarse time-scale), collecting information from long distance, is used to enrich token-level representations which only attend local neighbors within a fixed window. And the enriched token-level representations are attended by the decoder for summarization. **As we can see, the information flows of these models are rather different. Critically, our model finally relies on the fine resolution representations, while the other models depend on the coarse level representation or the combination of coarse and fine level representations.**
>
> Given the difference in modality, nature of problem, and information flow, we do consider our work as a significant contribution.
>
> If the reviewer is aware of any other related works, we would humbly request the paper references and we will gladly discuss their relations.
>
> 2. *Regarding description of the model:*
>
> We revised the descriptions according to your comments. The local self-attention per se is the same as used in prior efficient transformers like Longformer and BigBird. We added a detailed description and a visual illustration in the Supplementary B. We also revised Figure 1 and caption to make the top-down update clearer.
>
> 3. *Regarding experimental verification compute and memory efficiency:*
>
> We would like to clarify that we don’t claim improved efficiency over prior efficient transformers,  but we emphasize our efficiency over full-attention models like BART, which is one major purpose of the experiment on CNN-DM.  Our model saves ~36% training time (9 vs. 14 hours) and ~12% memory usage (23 vs. 26 GBs), compared to BART, given the same batch size (32), training epochs (5 epochs) , as trained on 4 A100 GPUs.  This memory save might not seem striking. This is because the maximum length of the documents are set to be 1024, to accommodate the maximum pretrained position embeddings of BART, making the full-attention model equivalent to a local attention model with window size 1024, which is not much larger than our model’s window size 256. However, as the document becomes longer, the memory usage difference is striking, considering that full attention models with long documents cannot fit into the GPU even with batch size 1.
>
>
> [1] Fan, H., Xiong, B., Mangalam, K., Li, Y., Yan, Z., Malik, J., & Feichtenhofer, C. (2021). Multiscale vision transformers. arXiv preprint arXiv:2104.11227.
>
> [2] Chen, C. F., Fan, Q., & Panda, R. (2021). Crossvit: Cross-attention multi-scale vision transformer for image classification. arXiv preprint arXiv:2103.14899.

---

### Official Review · Reviewer_Kugi · 2021-11-01

**Correctness:** 3
**Technical Novelty And Significance:** 2
**Empirical Novelty And Significance:** 2
**Recommendation:** 5
**Confidence:** 4

**Main Review:**


Strength:
- The hierarchical structure proposed in this paper is reasonable. By modeling the input document in a bottom-up and top-down manner, the model can model long documents in coarse and fine granularity levels.
- The performance on benchmark datasets looks good.

Weakness:
- In their paper, the author mentioned that the local attention window size is 1024, which is even larger than 512. I think such a large window size might not be local. However, in Table 1, most datasets are not that long and even shorter than 1000. Did the authors consider different window sizes? (except the window size in Table 3.). Also, how to slide the window across the inputs? Is there any overlap between local windows?
- The combination of top-down and bottom-up is new. But what's the performance without cross-attention? Also, it is not clear how the copy operation is implemented.
- The author mentioned that they use BART as the initialization. Did they consider initialization with vanilla BERT or just random? I think this might better show the effectiveness of the proposed method by removing the BART contribution.
- In the experiment part, what is the parameter for those baseline methods (BigBird, Longformer, etc.) Instead of only comparing with GPT-3 on model size, they should also report the model size for other models.
- The writing of this paper needs to be improved. Especially for the experiment part, what's the define and implementation of OracleAdaPool?  What is RL denote?

**Summary Of The Paper:**

Modeling long documents is a challenging problem. This paper works on this problem with top-down and bottom-up structures. The hierarchical structure proposed in this paper adopts local and top-down correction to make the model learn local and long-range dependency. The experiment on long document summarization benchmarks shows the effectiveness of their proposed method.

**Summary Of The Review:**

The top-down and bottom-up framework proposed in this paper achieves good performance on long document summarization tasks. However, despite the results, the novelty of this design is somehow limited since the bottom-up and top-down idea is not new. The experiment part of this paper is also not clear, some details are missing, and some issues need to resolve (please see the weakness part).

---

> ### Author Response · Authors · 2021-11-19
> **Response to feedback**
>
> Dear Reviewer:
>
> We highly appreciate your time and thoughtful reviews. We will address your concerns in order.
>
> 1. *Regarding the window size choice:*
>
> We chose to use a window size of 1024, following Longformer, which is most closely related to our model. Using the same window size of 1024 allows us to have a direct comparison with Longformer. Following your suggestions, we ablated the window size and summarized the results in the Supplementary C. The experiments show an effect of window size. That is, as the window size increases, the performance on all metrics enhances. The effect is quite large when the window size is increased from 32 to 256. The effect becomes smaller after 256, but the model performance can still benefit from larger window size. Please see the Supplementary C for more details.
>
> 2. *“in Table 1, most datasets are not that long and even shorter than 1000”*
>
> These datasets are standard benchmarks for long document summarization (except for CNN-DM). The numbers in the input length column (# Input Words) of Table 1 are the mean number of tokens. A model for long document processing should be able to cover the lengths of most documents. For example, the 90th percentile of document lengths for arXiv and PubMed are 16,108 and 7,234 respectively.
>
> 3. *“how to slide the window across the inputs? Is there any overlap between local windows?”*
>
> Suppose the window size is 4, then each token attends 2 tokens on its left and 2 tokens on its right. The attention window for adjacent tokens can have overlap. We add more details regarding local self-attention and a visual illustration in the Supplementary B.
>
> 4. *Regarding the performance without cross-attention:*
>
> The cross-attention between token and segment is the most critical operation of the method, as shown in the middle panel of Figure 1. Without this cross-attention, the summarization is based only on the bottom-up inferred representations, similar to prior efficient transformers. We did an ablation study, presented in the Supplementary C. We find that removing cross-attention leads to a clear performance degradation, indicating the effectiveness of the cross-attention-based top-down update.
>
> 5. *Regarding the copy operation:*
>
> As shown in Figure 1, after we obtain bottom-up-inferred token representations through local self-attention (left panel), these representations are used twice. First, they are pooled to initialize top-level representations (as shown at the Pooling part of the left panel). Second, they are used as queries in cross-attention to the top-level representations (as shown in the middle panel).
>
>
> 6. *Regarding model initialization:*
>
> Our method is flexible enough to leverage most pretrained encoder-decoder models. We choose BART because our major baseline, Longformer, uses the same initialization. Since the baseline is initialized from the exact same BART checkpoint, the contribution of our method to the improved performance won’t be confounded by the initialization. Also, other baselines are also initialized from some large-scale pre-training models. For instance, BigBird is initialized from Pegasus pre-training.
>
> Since our model is a seq2seq model, it is more natural to be initialized with encoder-decoder transformers. While it is possible to be initialized from BERT, it would then hardly be comparable to prior works.
>
> 7. *Regarding model size:*
>
> Following your suggestions, we added known model sizes (through open-source implementations or reports from other papers) to Tables in the Results section. Overall, our model has a similar number of parameters as other models except GPT-3.
>
> 8. *Regarding writing clarity:*
>
> We revised multiple parts of method and experiment descriptions to make them clearer and uploaded the revised paper. To answer your particular questions, OracleAdaPool means that the adaptive pooling weights are based on oracle reference summaries (Please see the blue words in the last paragraph of Section 2.3 ). RL indicates reinforcement learning. We don’t use any reinforcement learning methods. A baseline, GPT-3, employed RL for book summarization. Please refer to their paper [1] for detailed descriptions.
>
> [1] Wu, J., Ouyang, L., Ziegler, D. M., Stiennon, N., Lowe, R., Leike, J., & Christiano, P. (2021). Recursively summarizing books with human feedback. arXiv preprint arXiv:2109.10862.

---

### Official Review · Reviewer_VzGN · 2021-11-04

**Correctness:** 4
**Technical Novelty And Significance:** 2
**Empirical Novelty And Significance:** 3
**Recommendation:** 6
**Confidence:** 3

**Main Review:**

Strength
Even a little surprisingly, the performance is super good on all datasets. It beats all state-of-the-art models, including Longformer, LSH, BigBird, etc, by a decent margin (usually more than +1 R-1). The model is just initialized from BART. I feel it’s almost magic that fine-tuning a BART parameter based model could be that effective. A lot of prior work shares the same conceptual design principles (hierarchical, coarse-and-fine grained, local-global, etc.), but this model works very well indeed.

Weakness
I can’t really conclude the difference or principle which contributes to the performance gain. The authors developed a new way to assemble these blocks to build a model, and it empirically works well. However, are there any distilled piece we can take from this paper? What’s the killing component in this model? It’s not clear for me. Some ablation study would be helpful.

The paper is a very straightforward technical report, but it could be more thought-provoking or intellectual. I don’t mean to criticize the paper because it’s easy to follow. I sometimes prefer papers with more insights or thoughts, although I totally understand they are usually not accompanied with empirical success.

Minor:
I am not a huge fan of the usage of “inference” in this paper. All these “inference” modules are trained and fine-tuned. It is a little confusing when I read the first page.
The caption and presentation of Figure 1 could be improved. The caption contains too many terms which do not show in the Figure. Please consider unify the model name between the figure and caption.

**Summary Of The Paper:**

The paper proposes a new model architecture for abstractive text summarization. The framework assumes a hierarchical latent structure of a document where the top-level captures the long range dependency at a coarser time scale, and the bottom token level preserves the detail. The proposed model shows some advantage in both capability in capturing global and local context  and being computationally efficient.
The model is validated on many datasets and it beats almost all previous model.


**Summary Of The Review:**

It's a model architecture paper for neural abstractive text summarization. It works super well on many widely used datasets. Despite the empirical success, it lacks more principled insights and justification about what works behind it.

---

> ### Author Response · Authors · 2021-11-20
> **Response to feedback**
>
> Dear Reviewer:
>
> We appreciate your positive feedback on the empirical performance and your valuable reviews. Your comments are addressed as follows.
>
> 1. *Regarding ablation and take-away:*
>
> Following your suggestions, we conducted ablation studies and added them to the Supplementary C. The key component of our model is the top-down update to enrich the token level representations. Removing this component leads to a clear performance degradation. Furthermore, we consider a variant of the top-down update method. Instead of using token-segment cross-attention, we concatenate the token representations with the corresponding top-level segment representations. This change of the top-down update also leads to a decline in performance, indicating the good design of cross-attention for top-down update. Please see the Supplementary C for more details.
>
> To make top-down update effective, we argue that the initialization of the top-level representations is important. As shown in our experiments, top-level initialization with AdaPool (adaptive weights from a learned tagger) consistently leads to better performance compared to AvgPool (average pooling).
>
> Thus the take-home message from our work is simple: **the top-down update for token representations, especially with good top-level representations, leads to good summarization because of enriched token-level representations by the top-down**.
>
>
> 2. *Regarding the usage of inference:*
>
> We would like to clarify that inference here is in the sense of statistical inference (or posterior inference) for latent variables. Sorry about the confusion. We updated the paper to clarify this.
>
> We consider the token and segment representations as some latent variables (while they are not treated stochastically here). Computing the token and segment representations with the encoder is like some form of posterior inference given the observed text, and the encoder has a similar role as the inference network in variational autoencoder. Since the interaction of top-down and bottom-up representations in our work is inspired by the line of work investigating variational inference for hierarchical top-down generative models, we follow the terminology. This is also consistent with a recent theoretical analysis framework that links pretrained language models with an underlying latent variable generative model of text, where the downstream task is considered as computing a function on the posterior of the latent representations [1].
>
>
> 3. *Regarding the representation of Figure 1:*
>
> Thank you for your suggestions. We added terms from the caption to the figure. This indeed makes them align with each other better.
>
>
> [1] Wei, C., Xie, S. M., & Ma, T. (2021). Why Do Pretrained Language Models Help in Downstream Tasks? An Analysis of Head and Prompt Tuning. arXiv preprint arXiv:2106.09226.

---

### Public Comment · ~Peter_J._Liu1 · 2021-11-15
**Some clarifying questions**

Results look promising! I had some clarifying questions about the paper.

1. What are the maximum sequence lengths used for the tasks?
2. Is the local attention implemented with a sliding window (as in Longformer) or are they non-overlapping?
3. Is the segment-pooling non-overlapping?
4. Can you clarify which layer-norms are randomly initialized vs inherited?

Thanks.

---

> ### Author Response · Authors · 2021-11-24
> **Reply to clarifying questions**
>
> Thank you for your interest!
>
> We updated the paper to include all the details you mentioned. They're included in the Supplementary B.

---

### Decision · Program_Chairs · 2022-01-20

**Decision:**

Reject

**Comment:**

This paper deals with the task of long text summarization. Inspired by earlier work on top-down and bottom-up architectures, this work focuses on improving the traditional bottom-up converter encoder structure, and the fine resolution representations.

Pros:
- Their model can model longer documents in coarse and fine granularity levels.
- The performance on benchmark datasets looks pretty good compared to strong baselines
- Computationally efficient.

Cons: The reviewers have raised several concerns including:
- the experimental verification for calculation efficiency and memory usage of model is not sufficient.
- the novelty of this design is somehow limited since the bottom-up and top-down idea is not new.
- several details about the figures and especially the experiments were missing.

The authors have addressed several of the suggestions, added new experiments results addressing the issues raised by the reviewers. During the rebuttal period, the authors further conducted empirical investigations showing that the top-down update for token representations, especially with good top-level representations, leads to good summarization because of enriched token-level representations by the top-down. Despite positive results, some reviewers raised concerns that with only using BART as a backbone, it is surprising to achieve this great performance boost with the top-down/bottom-up models on long document summarization when they compared to the state-of-the-art transformer models (BigBird, Longformer and T5) that have been shown to encode longer sequences and beat several summarization models.